# Effects of Wearable Devices with Biofeedback on Biomechanical Performance of Running—A Systematic Review

**DOI:** 10.3390/s20226637

**Published:** 2020-11-19

**Authors:** Alexandra Giraldo-Pedroza, Winson Chiu-Chun Lee, Wing-Kai Lam, Robyn Coman, Gursel Alici

**Affiliations:** 1School of Mechanical, Materials, Mechatronic and Biomedical Engineering, Faculty of Engineering and Information Sciences, University of Wollongong, Wollongong, NSW 2522, Australia; jagp638@uowmail.edu.au (A.G.-P.); gursel@uow.edu.au (G.A.); 2Applied Mechatronics and Biomedical Engineering Research (AMBER) Group, University of Wollongong, Wollongong, NSW 2522, Australia; 3Department of Kinesiology, Shenyang Sport University, Shenyang 110102, China; 4Li Ning Sports Science Research Center, Beijing 101111, China; 5School of Health and Society, Faculty of Arts, Social Sciences & Humanities, University of Wollongong, Wollongong, NSW 2522, Australia; rcoman@uow.edu.au; 6ARC Centre of Excellence for Electromaterials Science, University of Wollongong Innovation Campus, North Wollongong, NSW 2500, Australia

**Keywords:** wearable device, running, biomechanics, performance, technique, biofeedback, gait retraining

## Abstract

This present review includes a systematic search for peer-reviewed articles published between March 2009 and March 2020 that evaluated the effects of wearable devices with biofeedback on the biomechanics of running. The included articles did not focus on physiological and metabolic metrics. Articles with patients, animals, orthoses, exoskeletons and virtual reality were not included. Following the PRISMA guidelines, 417 articles were first identified, and nineteen were selected following the removal of duplicates and articles which did not meet the inclusion criteria. Most reviewed articles reported a significant reduction in positive peak acceleration, which was found to be related to tibial stress fractures in running. Some previous studies provided biofeedback aiming to increase stride frequencies. They produced some positive effects on running, as they reduced vertical load in knee and ankle joints and vertical displacement of the body and increased knee flexion. Some other parameters, including contact ground time and speed, were fed back by wearable devices for running. Such devices reduced running time and increased swing phase time. This article reviews challenges in this area and suggests future studies can evaluate the long-term effects in running biomechanics produced by wearable devices with biofeedback.

## 1. Introduction

Running requires efficient conversion of power output into translocation [1], initiated by a greater joint range of motion [2,3]. During running, while the hip generates power to accelerate the leg so as to optimize the position of the foot and center of body mass [3,4,5,6], the ankle stabilizes and further accelerates the limb forwards [3,7,8], and the knee absorbs the loading by increasing the muscular power [3,9,10]. Biomechanics plays an important role in running performance and energy cost [7,11]. Attempts to change the biomechanical parameters of runners, characterized by quantitative spatiotemporal, kinematic and kinetic data [12,13], have been found to have some positive effects on running. For example, strategies to increase stride frequency (SF) and reduce vertical oscillation of the body as well as ground reaction forces allow efficient energy transfer [11,14,15,16]. Meanwhile, smaller dorsiflexion and faster plantarflexion are required [1,11,17] to achieve greater horizontal heel velocity and propulsion, knee flexion angle at initial contact (IC) [18,19] and greater maximum hip extension [20]. These biomechanical considerations are particularly important for competitive runners to decrease the completion time for a race distance and reduce the risk of injury [21]. Most runners are able to integrate and accommodate their own unique combination of anthropometric dimensions and mechanical characteristics to find a running motion, which is most economical for them [11].

Gait retraining has shown positive correlations in running mechanics, improving performance [22,23,24,25]. Biofeedback is a potential strategy to modify motor performance [25,26,27], although other strategies, including muscle strengthening, and training on explosive movements [28] and endurance, have also been used [29]. Although feedback and retraining are necessary to increase physiological and metabolic performances, they face challenges in obtaining accurate, quantitative, repeatable and continuous data about the athlete in the field [30]. This could explain why some previous studies suggested verbal feedback provided in the form of increments of intensity, frequency of the running, speed and time, which has resulted in low or no improvements in individual running mechanics [29,31] and require high manpower demand. There is also a possibility to underreport metrics related to injury, such as joint loading impact, because the human eye is unable to identify and quantify it [32]. Quantification of the biomechanical variables relies mainly on expensive laboratories set up [33].

Meanwhile, wearable technologies are options, which allow real-time measurements to be conducted in outdoor environments [34]. Wearables devices are lightweight mechanical or electronic technologies that are worn close to and/or on the surface of the skin. They are typically sensor-based devices that detect internal and/or external variables and transmit the information to an external device [35]. They usually come with some electronic components that provide auditory, visual or somatosensory feedback to the users [35,36]. In some cases, these devices analyze and transmit information immediately, generating biofeedback in real time [37,38]. Some latest techniques were developed to track balance [39,40], joint load [41], symmetry in the movement and joint angles [42]. They also gave instant tactile, visual and auditory feedback, which were found to be beneficial for the movement of a wide type of population such as healthy adults [43], amputees [44], older adults [45], runners [46] and stroke survivors [47]. Wearable sensors were validated against gait laboratory equipment on kinematic and kinetic measurements [48,49,50,51,52,53,54] in sports-related movements [55]. Accuracies of wearable sensors varied, as there were always challenges in the accurate position of sensors [56], data processing and simplification methods in calculating joint angles and segment acceleration [57] and protocols to calibrate the devices [58].

A systematic review of the current literature about the effectiveness of wearable devices with biofeedback in running biomechanics is needed. A previous review has looked into wearables devices without real-time feedback in sports players, including runners [59]. Another review [22] undertook a narrative review of psychometric parameters such as motivation for running, preferred characteristics about feedback platforms, content feedback programs including frequency, motor learning strategies and workload configurations. However, these reviews did not focus on biomechanical parameters such as vertical reaction forces and running kinematics.

The primary aim of this systematic review is to determine whether wearable devices with biofeedback are able to modify and impact the biomechanics of the running and its performance. To achieve this, the most key parameters that were used to analyze and modify running gait were identified, and the results from previous studies with the use of wearable devices with biofeedback programs were organized. We hypothesize that wearable devices providing real-time biofeedback would produce biomechanical changes in the running technique and thereby improve performance.

## 2. Materials and Methods

A systematic literature search and analysis was conducted and reported in line with the preferred reporting items for systematic reviews and meta-analysis (PRISMA) guidelines.

### 2.1. Study Inclusion and Exclusion Criteria

This systematic review includes articles that focus on wearable technologies providing feedback on biomechanical parameters related to running. The search was limited to peer-reviewed journal articles written in English and published in the period from March 2009 to March 2020. Records published before 2009 were searched but not retrieved. Book chapters and conference papers were excluded.

The articles include wearable devices that provided feedback to users regarding kinetic, kinematic or spatiotemporal variables of running and should report results related to the sports gesture and performance in healthy adults. Articles were excluded if they included patients, animals, robotic-assistive devices, orthoses, exoskeletons or virtual reality environments. Articles were also excluded if their primary outcome measures were related to heart rate, sweat, sleep or cognitive/emotional conditions, classification/recognition of patterns, level of physical activity or navigation.

### 2.2. Search Strategy and Study Selection

The review process was completed in four steps. First, potentially relevant records were identified through a systematic search for published papers in four major scientific databases of Scopus, Cochrane, EBSCOHOST (SportDiscus, CINAHL, MEDLINE AND PUBMED) and Web of Science databases.

The search strategy involved a combination of keywords and subject headings terms presented in Table 1. The simplified strategy was (1 or 2) and 3 and (4 or 5 or 6 or 7) and (8 or 9) and not 10. According to the restrictions in each database, the identifier was searched by sections before being included in the main search strategy.

During the second phase, the title and abstract of all records were screened for relevance. Full-text articles were retrieved and assessed if the relevance was unclear. For the third phase, all records not fulfilling the inclusion criteria were excluded. Eligibility was assessed for each remaining full-text article. Finally, relevant information was extracted from the included records. References of all included studies were checked for additional publications that could be included in this review.

### 2.3. Data Extraction

The author who did the search extracted data from each included article. Information was compiled in separate tables from each article regarding the participants, protocol, sensor(s), feedback and analysis. Demographic information, type and location of the sensors, feedback devices, type of feedback, type of environment, length of protocols, length of interventions as well as each biomechanical variable were analyzed in each record. Means and standard deviations for outcome measures were also extracted from all included articles for data analyses. Further comparisons were made between protocols, hardware device characteristics, feedback modalities and effects of biofeedback on running performance.

## 3. Results

### 3.1. Search Results

A total of 417 articles were identified through the database search, and 5 additional articles were identified through a manual search from reference lists. Following the removal of duplicates, 375 articles remained. Screening of the titles and abstracts led to the removal of 256 articles. Upon assessing the full texts of the remaining 119 articles, 19 articles were found to have met the inclusion criteria (Table 2, Figure 1). A list of the abbreviations used in this manuscript and their definitions is giving at the end of this manuscript.

### 3.2. Subjects

The participants in the nineteen reviewed articles were healthy recreational or competition-oriented runners in both genders aged between 18 and 47 years with a median age of 26 years old. They ran a minimum distance of 10 to 17 km per week. Some studies recruited runners with a high-risk running technique [60,61,62] or a peak positive acceleration (PPA) higher than 8 g [63,64,65,66]. One study [67] included overweight children as participants in their experiments (Table 2).

### 3.3. Components of Wearable Biofeedback Devices

#### 3.3.1. Use of Wearables Sensors

The most popular sensors used to evaluate running performance were accelerometers [56,61,62,63,64,65,66,67,68,69,70,71,72,73,74]. Among fifteen included articles using accelerometers, three used wired accelerometers [61,66,71], six articles used wireless technology [62,63,67,70,72,73] and the type of connection is not mentioned in the remaining six articles [56,64,65,68,69,74]. The number of axes in the accelerometers varied, including uniaxial [68], biaxial [61,63], triaxial [62,64,65,69,70,71,72,73,74]. Some studies used inertial motion units (IMU), which contained triaxial accelerometers together with triaxial magnetometers and gyroscopes [67,73]. Two included articles did not disclose the number of axes of their accelerometers [56,66], and in another two articles, the accelerometer was part of commercial devices such as Aximo PADIS [73] and Garmin FR70 [62,72]. The measurement range of the wireless accelerometers varied between ± 2.8 g to ± 50 g, whereas the laboratory-based did not include their range in their description.

The sampling frequency (SF) of the accelerometers ranged between 100 and 500 [62,67,68], ≥500 [63,70] and a majority over 1000 Hz [56,61,64,66,69,71,72,73,74,75]. The sampling frequency of accelerometers for the rest of the papers was not mentioned.

The most popular body place among studies that used accelerometers was in the anteromedial aspect of the distal tibia, approximately 15 cm above the medial malleolus [56,61,63,64,65,66,67,68,69,70,74]. Two included articles placed accelerometers on the head [56,74] and at the top [62,72,73] and back [61,64,71] of the shoe (see Figure 2).

There was one study [75] using force resistors and capacitive sensors, which were embedded in an insole. The maximum load for each of these sensors was not specified. The force sensor resistors were used as footswitches (MA-153 Motion Lab System), while the capacitive sensors were part of an instrumented insole sampling at 100 Hz with a telemetry-based system.

One study used a wearable position transducer [76], in which a waist strap was placed on the back of the runner, and the vertical displacement was measured (see Table 3). Two included studies [60,77] collected data from motion capture systems, which were not wearable, but did they have some wearable devices which provided wearable biofeedback.

#### 3.3.2. Use of Feedback Modalities

The most common type of feedback (twelve articles) among the reviewed articles was visual [56,61,62,63,66,67,68,69,71,72,73,74]. Two controlled trials used auditory cues [64,70], two haptic [60,77] and the remaining articles mixed at least two types of feedback [65,69,75,76].

Within the visual modality, the most popular form of delivery was a screen on a monitor placed horizontally to the head, in front of the treadmill except by Busa, who projected the head gaze on a screen of 1.2 × 1.6 m [74]. The monitors projected lines, dots, colors and shaded zones suggesting specific tibial acceleration, area of trajectory or foot loads targets [61,66,67,68,69,71]. Another form of visual delivery was by screens on watches attached to the wrist or screen on phones with biological data [62,72,73,75].

The auditory feedback was delivered through verbal instructions [56,69,75,76], sound without modulation [70,77] or sonification based on error or performance [64,76]. One study used a wireless headset with pre-recorded messages with different tones according to the error magnitude [76], and the rest of the articles presented cues on speakers (Table 3).

The haptic device involved a bodysuit delivering permanent body compressions [77] and a portable device creating a 5 kg resistance on the legs that limited knee flexion and perception during the run [60]. None of the articles used vibrations as feedback.

#### 3.3.3. Instructions to Participants

Before starting the trials, participants were usually given verbal instructions about the expected cues they were about to receive and their meaning. If the feedback was displayed on a screen, participants were instructed to land softer [61,63,65,66,67,68,70,71,74,76], land with a greater flat foot [75] or reduce the foot eversion angle [56] in order to maintain the signals within the targets. If the feedback was auditory, some studies asked participants to follow what the voice message suggested to do, and some studies explained the meaning of different pitches or tones before the trial. For haptic stimuli, participants did not must follow any instruction on what to do about the sensation, but they were instructed to run as normal while permanent compressions around the body were applied (Table 3).

### 3.4. Outcomes Measures of Running

#### 3.4.1. Equipment

All included articles used laboratory-based equipment to collect gait data and investigate the effects of biofeedback on running. Eight articles used treadmills, together with wearable accelerometers, to measure the outcomes of feedback. Six of those used treadmills instrumented with force plates [61,63,64,66,68,71] and two used regular treadmill without force/pressure measurements [67,70]. The protocols that used a non-instrumented treadmill limited their analysis to only tibia acceleration [67,70], while the articles that used an instrumented treadmill allowed studies of ground reaction forces [61,63,64,66,68,71].

Eleven included articles studied changes in running gait utilizing a motion analysis system. Motion capture systems including Qualysis [62,65,70,72,74], Vicon [56,60,66,69] and reflective object sensors from Optek Technology [77] were used.

Other studies combined data collection with force and position transducers [68,76], and two articles used equipment to measure physiological parameters, including heart rate with straps [73], VO2 max with mouthpiece and piezoelectric electronics (PhysioDyne) [56] and lactate levels from capillary blood sampling [73] (Table 3). Five articles used commercialized wearable devices, which were not part of the wearable technologies and aimed to improve running mechanics and/or performance [62,67,72,73,74].

#### 3.4.2. Study Environment

The environment for the studies was either indoors [56,60,61,63,64,65,66,67,68,69,70,71,74,75,76,77] or outdoor protocols, mixed with indoor gait assessments [62,72,73]. Indoor protocols were required to track and control signals in laboratories, either for measuring or delivering feedback signals. The Indoor assessment involved the use of an instrumented or a non-instrumented treadmill, which allowed analysis of running over short distances. If the protocol was outdoor, participants ran their normal training [62,72], allowing long-distance running to be conducted. One included article [77] analyzed sprints as well as running; some other articles analyzed running over inclinations [73] and simulated irregular terrains [56]. The speed of the protocols was either not controlled (e.g., during a training session) [62,72,73], controlled based on a self-selected pace with a range between 6 km to 20 km·h^−1^, the manipulation was made by increasing 10 to 20% the baseline [63,64,67], controlled by other gait parameters such as cadence or step rate [62,72,74,76] or by a set pace (e.g., 14 km/h) [75].

### 3.5. Effects of the Use of Biofeedback on Running Performance

A variety of training programs were found across the articles. Table 4 presents a summary of each protocol and the conditions of each gait variable that suggested changes in the gait pattern.

#### 3.5.1. Peak Positive Acceleration (PPA)

Biofeedback for PPA: Ten included articles measured the tibia or shoe PPA during running at the self-selected speed [61,63,64,65,66,67,68,69,70,71]. They examined the highest peak of the acceleration that occurred during the early stance phase of running, then the data processed according to the number of axes measured in a customized Matlab or Labview program that each author built. They set the PPA values as the baseline before the use of biofeedback. Four articles used 50% reduction of the PPA as a variable [65,66,68,69], four articles used 20% [61,63,64,71] and one article used between 15% to 20% [70] of the baselines as thresholds for biofeedback.

Biofeedback was triggered if the real-time measured PPA was above the thresholds. Three articles used two thresholds, combining a 20% reduction of the PPA with additional factors of a cognitive task [64,71] and running speed [63]. One article combined speed, cognitive task and a self-selected PPA reduction [67]. Five articles provided the biofeedback constantly on monitors in graphics or numbers [63,66,67,68,69], whereas the remaining five articles presented binary signals through colors on a screen [61,65,71] or auditory signals such a beeping or pitch [64,70] biofeedback only when the targeted values were not reached.

From the ten included articles that fed back PPA to modify the gait, five [61,63,64,66,71] followed the same indoor protocol suggested by Crowell et al. [66]. This consisted of a retraining program where participants ran following suggested changes. Those changes were given based on the collected data from wearable devices or from a laboratory setup. The program had eight sessions with the run time increasing gradually from 15 to 30 min, and the feedback time was fade from the fourth session. The fade strategy outlined reducing the time from 30 min to 1 min across sessions during the last session; the feedback would last a minute. The remaining articles ran six [65], two [69] and a single [68,70] session of 20–30 min, except by one article [67] that included eight short running sessions. An article followed up the changes 7 days after the end of the retraining program [69] and three articles [65,66,67] after a month. These sessions were called retention sessions.

Biomechanical effects for PPA (Table 5): PPA was significantly reduced upon using the biofeedback by 10–20% [67,70], 20–30% [61,69], 30–40% [63,64,65] and 40–50% [66,71]. Many articles additionally studied vertical instantaneous loading rate (VILR) and vertical average loading rate (VALR). VILR and VALR were calculated from the most linear portion of the vertical ground reaction force curve during the early stance phase. The VILR corresponds to the slope of the line from 20% to 80% of the impact peak, whereas the VALR corresponds to the maximum slope between adjacent data points in the same region. VALR was also significantly decreased by 15–20% [61,65], 20–30% [64,67,71] and 32% [66]. For the case of VILR, significant reductions were found between 15% and 20% [61,65,71], 22% [64] and 34% [66]. One article [68] reported a reduction in these three parameters, although the statistical difference was not achieved.

The articles that fed back both PPA and cognitive load [64,71] demonstrated a reduction of more than 30% of the PPA and more than 20% in VALR and VILR, however article [64] reported an only significant correlation between the biofeedback and changes in PPA, but not in ground reaction forces (GRF). Three articles [64,69,71] presented additional data related to immediate biomechanical changes on the run while the participants were still receiving feedback, whereas the rest only reported changes after the retraining sessions had ended (no feedback on). They reduced PPA by 29%, 26%, 22.3%, respectively.

Two articles [64,71] reported statistically significant reductions in VALR (9.7% and 17%) and VILR (7.6% and 13%) that appeared immediately upon using the biofeedback. Five articles reported changes after the retraining session (one week and one month after the gait retraining) [63,65,66,67,69], which proved that the changes in PPA remained with similar values to those ones obtained after the feedback was no longer active (post-training). [65,66] also followed up on VALR and VILR and reported similar findings; at least 5% and close to 30% of reductions, respectively. By manipulating PPA, one article also reported significant reductions in foot strike angle [65].

#### 3.5.2. Stride Frequencies (SF)

Biofeedback for (SF): The biofeedback was delivered mainly through signals displayed on an external device available to be taken outdoors [62,72] or on a monitor combined with sounds and commands requesting changes in stride frequency [74,75,76]. With visual and auditory feedback, metronome [74,75], a monitor showing the foot contact area [75] or head trajectory [74], commands and error tones from headphones [76] were used.

Baseline stride frequency was measured at a self-selected cadence. Thresholds for biofeedback were defined as 90% [76] (38), 100% [76], 107.5% [62,72], 110% [74,75,76] or 120% [74,76] of the baseline cadence. In some studies, SF thresholds were combined with foot load area [75] and vertical body displacement [76]. The biofeedback signal was produced once real-time measurements on stride frequency were below the threshold [74,76]. Two articles had the biofeedback signal remained on from the beginning of the session [62,72].

Two studies gave runner eight sessions of using the biofeedback over four [72] and two [62] weeks. They were conducted outdoors; however, all subsequent data collection was made indoors. The length of the run varied according to the physical capability of the participants. They were able to self-control how often they would rely on the biofeedback signal given by the devices. However, Willy [62] restricted the feedback to specific sessions. The remaining articles [74,75,76] performed single retraining sessions. The feedback time varied, a study [75] used five minutes, and the outcomes were measured during the last 30 s of each trial. Two articles [74,76] did not state the duration of the use of biofeedback but rather stated the number of trials where biofeedback was provided.

Biomechanical effects for SF (Table 6): Five included articles provided feedback on stride frequency of the run [62,72,74,75,76]; however, the variables analyzed across all articles were not homogeneous. They could be widely grouped into foot [62,72,74,75], knee [62,72] and hip [72] modifications; changes in the center of pressure (COP) [62,75], vertical displacement [76] and step length [62]. Comparisons regarding protocol schemes and results are difficult as the variables were different (Table 6).

All articles reported some biomechanical changes when biofeedback was used to remind users to increase SF of at least 7%. Willy [72] reported a decrease in ground reaction forces (VALR and VILR) by 18%, whereas Busa [74] reported an increase in tibia shock attenuation by 12%. Shock attenuation was defined as the gain during a cycle instead of only the peak. Phanpho [75] found such biofeedback, combined with foot loading suggestions, decreased 80% of heel region load, increased 10% the forefoot load and 45% forward displacement of COP during stance phase were produced. Interestingly Willy [62] reduced 22% the foot strike angle by increasing stride frequency, whereas Clansey [65] reduced it by 45% together with a reduction of 50% of PPA. There was at least 10% knee load reduction [62,72], 25% increment of knee flexion at foot strike, 27% reduction in hip adduction [72] and 9% reduction in COP movement [62].

Eriksson [76] demonstrated that with minimum changes in SF, the vertical displacement could be reduced up to 20%. However, when comparing fed back SF through visual or auditory cues, it seems that participants were able to achieve running targets easily when auditory cueing was presented [76]. Gait data in most studies attempting to modify SF was taken after the retraining session was finished, with the exception of a study [62], which made an additional full run analysis investigating immediate changes upon using the biofeedback. That study decreased 5% COP movement, 7% the step length and 11% the contact knee forces. Additionally, kinetic and kinematic changes lasted significantly for one-month in two articles that structured the biofeedback across eight sessions [62,72].

#### 3.5.3. Other Parameters

Other parameters that were manipulated by wearable devices with biofeedback included contact ground time (CGT) [73], foot inversion angle at heel strike [56], speed [77], and the resistance applied to the lower legs during the run [77]. The protocols and parameters to suggest run changes are different; however, they have similarities with other articles among the variables that they modified. (Table 7).

Among four included articles analyzed in this section, three articles [56,73,77] did additional metabolic measurements such as heart rate [73,77], maximal oxygen uptake (VO2max) [56], tissue oxygenation index (TOI) and lactate [77]. They reported decrements or no changes in the physiological response, suggesting that the metabolic demand did not reach levels of fatigue, but biomechanical changes were observed.

A single article modified mechanics in the run from modifying spatiotemporal parameters [73]. The researchers did not select a specific threshold, but rather they asked the participants to decrease the known contact ground time from the last run (CGT). Results showed there were no significant reductions in CGT; however, those small changes produced a significantly longer swing time. They reported a significant reduction in the SF and running time of 1.5% without increasing the metabolic demand.

One individual article provided biofeedback on speed [77]. The thresholds were based on the self-selected sprint peak, and six target speeds were tested (20%, 35%, 50%, 70%, 100%), being 50% the value that represented running. The feedback signal was given by a monitor with the target speed, and a stretchable textile full-body suit continuously compressed the body regardless of the speed target. This was the only article that increased the distance covered (7%), and likewise, Zhang [63] reported similar increments in the speed (8.5% vs. 10%). Three other articles included changes in the speed combining PPA values. The feedback strategy and the threshold definition were mentioned in the PPA section [63,64,67].

Another article provided feedback on the frontal plane foot angle (inversion) at heel strike [56], which was able to decrease 14% of the inversion with no alterations in the shock attenuation. The threshold was a preset measurement of the self-selected inversion foot contact angle when participants ran on a man-made irregular surface. The retraining session consisted of running over a smooth surface with the preset foot angle as a target. The biofeedback signal delivered a continuous visual gray area on a monitor, and participants aimed to fall into that area. As a result, an increment of more than 15% in knee flexion was reported. This finding is aligned with the results in the article [62] however, results in article [56] need to be read carefully when comparing with other articles, as the control group runs on an irregular surface rather than a smooth surface.

Although articles [62,65] analyzed foot kinematics after SF and PPA were manipulated, the only included study that provided feedback on foot kinematics was article [56]. Comparison between foot changes across articles is impossible as the variables measured were different (inversion vs. dorsiflexion).

One included article [60] did not provide feedback from a gait baseline measurement nor generated measurements informing about gait parameters. Participants wore a belt with bands reaching the lower leg. The belt provided haptic feedback by applying 5 kg of resistance during the run. Gait data collection and analysis was made after the belt was removed from the body. This protocol decreased the vertical displacement by 12.7%, increased 6% the knee flexion and reduced by 34.7% the vertical angle of the shank.

Articles grouped in this section reduced the vertical tibial angle with a flatter foot [60], decreased the metabolic cost [56,73,77] and decreased the vertical oscillation [60]. These results may suggest that the internal work may also be reduced. However, the protocols among these articles were designed to produce changes within a session. There is no information available about the long-term effects of the feedback. Therefore, it would be beneficial to evaluate the findings of longer time-interventions.

## 4. Discussion

Vertical loading, which is positively correlated with peak tibia acceleration [78,79], was most frequently used in the included articles to investigate the effects of wearable devices on running. Approximately 10% incidence of injuries are related to tibial stress fractures in runners [68,80], and there is a recurrence rate of 40% [81]. Tibial acceleration, VALR and VILR are good indicators for risk tibial stress fractures [82] and, therefore, key variables in running injury prevention. While there is a lack of support in the literature associating these variables with an ideal running technique that improves the running time, speed and economy [11,83], they have an indirect effect on the running performance through vertical loading and injury prevention.

The data presented across all articles shows that biofeedback can reduce variables related to GRF in the short term, including not only the PPA as an indicator of the impact force but also the vertical loading rates. Crowell [66] achieved the highest GRF reduction among the studied articles by providing visual feedback on the PPA. Their changes remained after a month, but their protocols were not randomized. In contrast, Clansey [65] randomized and increased to double their sample size, producing a more robust protocol than Crowell [66]; however, the changes in PPA were smaller than those found by Crowell [66] at a one-month time (20% vs. 44%).

The VIRL and VALR were also found to be reduced at least 15% across articles that modified solely PPA. However, at the one-month time, changes remained in a small proportion (over 5% reduction) and were not able to be maintained [65]. This suggests that changes need to be made to promote robust retaining values for longer periods of time. It appears that subjects were able to reduce values in PPA, VILR and VALR while the biofeedback was present. After the biofeedback was removed for a short or long time, they were unable to maintain the same levels of reduction; however, they kept the values lower than the controls [64,65,66,69,71]. It is interesting to note that those articles using a faded biofeedback protocol achieved higher values than those that used constant feedback

Wearable biofeedback devices that modified simultaneously speed and PPA are able to decrease GRF in a distracting run [64,71]. With additional cognitive load, visual feedback [71] produced higher reductions on PPA than the auditory feedback [64], and both produced similar changes in GRF. They reported similar values to those modifying only PPA [69,70]. However, they have different protocols, population and PPA targets. Therefore, comparisons about best practices need additional studies.

The literature suggests that a shorter contact time will represent a shorter breaking phase [84]. Findings in one analyzed article suggest that small changes in the ground contact time increased the performance by reducing the time that runners spent covering the same distance without increasing the metabolic response [73]. These results are supported by Clark [83], who argues that there is not a strong statistical correlation between running performance and the magnitude of the GRF, but with the contact time. The provided visual feedback in article [73] was not permanent. However, the authors suggest that participants may have difficulties judging the magnitude of the CGT while they were running; therefore, implement changes was challenging. Major conclusions related to GCT and running performance cannot be drawn.

Although running time and mileage are two important factors during the run [1], only two articles reported improvements of those parameters from wearable devices. However, the evidence about the effect of the biofeedback from wearable devices on these variables and the relationship with vertical loading and lower limb kinematics is not yet conclusive.

While step length and frequency are mutually dependent, and their balance defines running speed and running time [11,21], there has been little quantitative analysis among wearable devices of step frequency related to running time. The results proposed that, even when the reviewed articles aimed to increase the running performance, their main focus was on the biomechanical variables that may prevent injury instead of the spatiotemporal parameters [62] that may increase running mechanics. Further exploration needs to be done, identifying if the strategies that reduce overloading allow the individual to run faster with no extra metabolic cost.

Trained runners tend to have higher stride frequencies than untrained runners, but a self-optimization seems to be the most efficient path to improve performance [1,21]. Runners have shown a preferred optimal stride frequency that minimizes metabolic cost due to increments in the power from elastic energy reducing the absolute muscle power [85]. In contrast, no articles analyzed the metabolic response when modifications were made in SF except by an article that suggests that increments above 10% in SF is metabolically costly, but small changes may be enough to reduce the loading rates [72]. Additionally, to determine best practices related to SF, more literature is required as in terms of variables analyzed, the articles included had in common only the SF [62,72,74,75,76].

Elite athletes have less vertical oscillation than recreational runners, suggesting that reduction in the COP excursion or COM displacement represents higher performance [1,86]. Conversely, reductions in the COP excursion and vertical displacement reduce heel load and shank anterior muscles work unless the runner presents excessive dorsiflexion or foot inversion [87]. The results from the analyzed articles are according to the literature [62,75]. Interestingly, when runners are given the option to modify either vertical displacement or SF to achieve targets suggested from wearables, it seems they prefer to make adjustments in the vertical displacement that changes in the SF [76].

Changes were exhibited in the analyzed articles in foot, knee and hip by providing biofeedback on SF from wearable devices [62,72,74,75]; those changes are in agreement with the current literature that suggests that discrete increments in SF lead to kinetic changes and are beneficial for preventing injuries while reducing the lower limb loading and impact [19]. Greater forefoot strike reduces vertical forces and, consequently, the vertical rates [87,88]. Additionally, misalignments increase internal joint work and would negatively impact the running performance [88]. Likewise, a reduced plantarflexion at heel strike, actioned by the tibia, prevents injury because the contact area of the knee increases. Therefore the joint stress would decrease [87,88,89].

Better running performance has been associated with a lower percentage of stance phase, smaller peak breaking velocity of the pelvis, greater upright trunk [90], slower velocity in the knee during stance phase, faster knee flexion velocity during swing phase as well as faster hip flexion velocity between heel strike from opposite feet [90,91,92]. These parameters were not involved in any of the articles analyzed.

This review found large variations among studies in variables that were analyzed during the retention sessions (from one week to one month after removing any biofeedback) as well as the post-training test (immediately after feedback was delivered). One possible explanation for such a difference may be related to motor learning and the way how the feedback was delivered. Long term effects in the running technique rely on the motor learning process. A key concept among motor learning is the capacity to transfer learning skills to different environments by relying on intrinsic biofeedback signals that are built upon previous training. In a retraining session, it is expected that the intrinsic feedback adjusts accordingly to the stimuli provided by the biofeedback from the wearable devices. Motor learning process literature has proven that reducing the feedback signal would gradually increase the demand for intrinsic feedback, as well as the possibility of retaining gait changes after long periods of time without an external feedback signal [93]. This may be the reason why during the retention session, article [66] maintained a significantly lower PPA, and even when article [72] produced changes from SF and not from PPA, they also produced significantly lower levels of VALR and VILR than article [65]. More conclusions require further research, as only four articles analyzed the changes in one-month time [65,66,67,72].

Direct comparisons between the efficiency of the biofeedback modalities among articles that modified SF over foot strike angle and vertical loading are not possible. However, in outdoor environments, the manipulation of SF with wearable devices through self-controlled visual biofeedback [62,72] provided similar changes to the PPA-laboratory-based visual biofeedback [65]. The auditory feedback seems to be the most efficient modality when modifying the foot load distribution indoors with the potential of transferring the protocols outdoors [74].

A conclusive biofeedback modality that can be used to improve running technique still needs to be further analyzed. Human responds effectively to biofeedback signals under dual or single task during the run. Visual feedback is an effective strategy to provide signals about the performance of the run in real time indoors and outdoors [61,62,63,65,66,67,68,69,71,72,73,74,75,76,77], and subjects prefer signals that contain significant information rather than numerical values. Auditory biofeedback also generated significant gait changes when the PPA and SF were the parameters leading the changes with a lower number of articles [56,64,70,74,75,76].

All included articles that used auditory or visual feedback included electronic wearable devices. Haptic feedback was used only with mechanical wearable devices that were not compounded by electronics [60,77]. There is a gap in the literature that uses wearable haptic technology producing modifications in running technique. Further exploration is needed from both electronic and non-electronic devices in order to determine its full potential. It should also be noted that biofeedback may compete with the primary motor task for working memory resources. If the subject’s motor control depends largely on working memory for conscious processing, biofeedback is likely to limit the available attention resources and thereby affect running mechanics. Further research can devise biofeedback modalities that allow athletes to focus on multiple tasks and narrow programs to different populations.

**Future research direction: **Further research in the running efficiency and economy upon using wearable devices with biofeedback would be valuable. Wearable devices are a powerful tool to modify the running technique, yet, laboratory equipment was used to evaluate the effectiveness of wearable devices. Further research on validation and calibration of these devices would be important. More studies should investigate interventions in the field to fully understand the effects of such technology on running outdoors.

Long term effect upon interventions with wearable devices is an unexplored area. Research has focused on short terms effects upon short interventions. However, the evidence highlights promising long term effect on the positive peak acceleration in those protocols where the subjects were exposed to longer periods of time, and the task did not focus on their metrics. It would be beneficial to increase the number of clinical trials exploring the long term effects of wearable devices with multiple biofeedback modalities in biomechanical parameters.

**Limitation of this systematic review:** First, we limited our search to peer-reviewed journal articles. There may be some other protocols and strategies used to modify running techniques mentioned in other types of publications. However, peer-reviewed articles usually have better protocols and larger sample size.

Second, we also excluded studies that involved the validation of new sensing technology purely for running. They may have included some strategies to improve running performance and tested them with a very small number of participants. Nevertheless, these studies are usually unable to provide enough evidence to support the benefits of such new technologies on running.

Third, articles that included orthosis and prosthesis in their keywords were excluded. It is possible that technology such as haptic devices embedded in orthosis such as insoles may have been excluded. However, they tend to be associated with strategies among a pathology rather than with running performance, or they may modify the initial mechanics by changing the basal running position involuntarily.

Fourth, the results should not be extrapolated to a different population than healthy runners. It may also be beneficial for those runners that are at risk of a tibia fracture and overweight children. There is no data about elite runners. Therefore although this review may benefit them, further analysis needs to be made in this type of population to suggest best practices.

## 5. Conclusions

The purpose of the current review is to determine if wearable devices with biofeedback increase running performance. It has shown that, according to the available literature, positive peak acceleration, which was found to be associated with a tibial stress fracture, was significantly reduced upon receiving its biofeedback. By providing biofeedback to increase stride frequency, participants had significant reductions in the knee and ankle vertical load and a significant increase in knee flexion. Positive changes were also found in wearable devices, which fed back other parameters (contact ground time and speed), as they reduced running time, vertical displacement and increased greater knee flexion. Taken together, these results highlight the possibility of improving running performance and injury prevention.

Auditory and visual biofeedback have been proven to promote changes, mainly in the short term. Haptic feedback successfully improved performance in the short term. However, more research needs to be done as only two included articles utilized haptic biofeedback for gait retraining. Interventions where the participant does not fully depend on the biofeedback system, task-oriented commands and longer protocols over longer period of time, are recommended in future studies. In addition, more randomized-controlled trials that are conducted in outdoor environments are required to better evaluate the effects of wearable devices on runners with different levels.

## Figures and Tables

**Figure 1 sensors-20-06637-f001:**
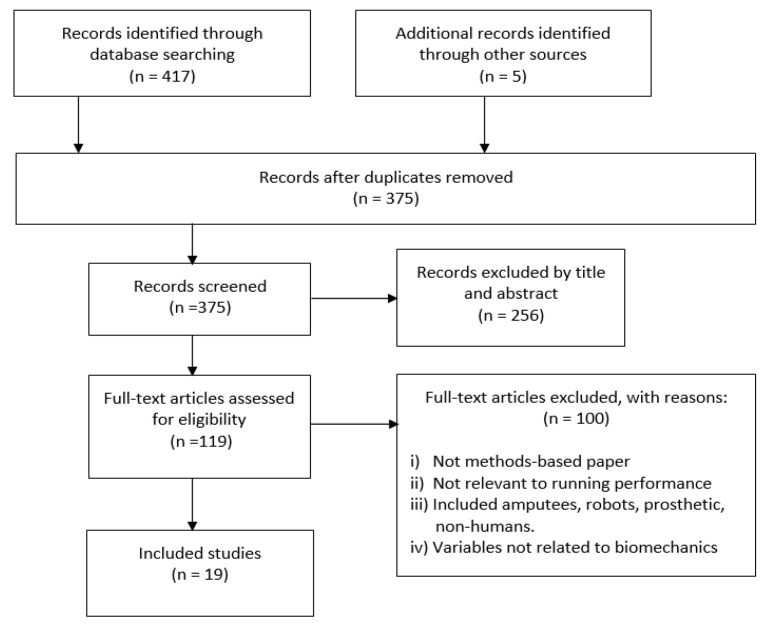
Flow diagram of study selection.

**Figure 2 sensors-20-06637-f002:**
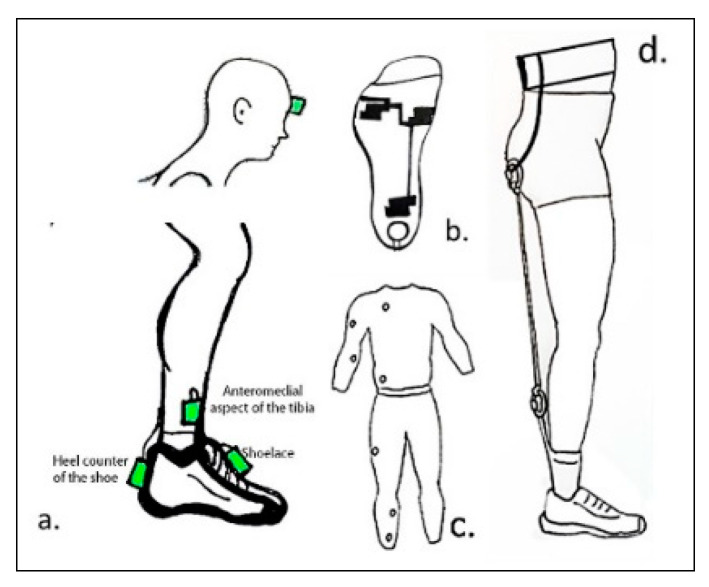
Schematic of wearable devices. (**a**) Representation of accelerometers and IMU placed in one or two body parts; (**b**) a heel-force resistive sensor and three embedded conductive sensors were placed in a sock connected; (**c**) a full-body compressive suit; (**d**) running belt device applying 5 kg resistance.

**Table 1 sensors-20-06637-t001:** Keywords and subject headings descriptors grouped by means.

Terms and Strategies	Identifier
Wear * (technolog * or device * or sensor *) or “real-time sensor”	1
Inertial sensor” or “inertial measurement unit” or gyroscope * or magnetometer * or acceleromet * or “cell phone” or “smart phone *”	2
Run * or running or “runner” or “running injuries”	3
Feedback * or biofeedback * or augment * or “real time”	4
Visual (signal * or feedback * or cue or biofeed * or augment *) or (visual (train * or retrain *))	5
Auditory (signal * or feedback * or cue * or biofeed * or augment *) or (auditory (train * or retrain *))	6
Haptic (signal * or feedback * or cue * or biofeed * or augment *)) or (vibrat * (signal * or feedback * or cue or biofeed * or augment *)) or (haptic (train * or retrain *)) or (vibrat * (train * or retrain *))	7
Mechanic * or load * or performance or postural * or “ambulatory monitoring”	8
“Ground reaction” or force or kinematic * or biomech * or acceleration * or cadence or “step length” or “step width” or “step time” or “stride length” or “stride time” or “stance phase” or “swing phase” or “stance time” or “swing time” or “single support” or “double support” or “ground contact” or “gait speed” or “walking speed” or “running speed” or “heel-strike” or “toe off” or “speed” or “center of mass” or “center of mass” or “center of gravity (CG)” or “center of gravity”	9
Patient or stroke or Parkinson’s	10

(*) used as a truncation command, searching for the root of the word and retrieving any alternate ending.

**Table 2 sensors-20-06637-t002:** Study identification and sample characteristics.

Citation	Year	Sample	Participants Conditions	Age (yrs)
[75]	2019	15:11 F; 4 M	Healthy recreational runners. Run: at least twice a week for at least 30 min per session. Rearfoot striker pattern.	25.67 ± 3.99
[60]	2019	10:7 F; 3 M	Healthy runners. Run: at least 10 miles per week. No injuries in the last 3 months, at least 2 factors of high-risk running technique.	28.3 ± 2.8
[63]	2019	13:3 F; 10 M	Healthy recreational runners. No injuries. Run: weekly running distance of 30.7 ± 22.2 km. Running experience: min 2 years (6.8 ± 4.4 yr). PPA tibia > 8 g.	41 ± 6.9
[67]	2019	37	25 over-weight children (11 CG, 14 IG). 12 non weight children (CG)	9.05 ± 1.64
[61]	2019	14:7 F; 7 M	Healthy recreational runners. Run: more than 12 km per week. No injuries in the last 12 months Rearfoot strike pattern. Foot strike angle > 8°.	26 ± 11.2
[56]	2018	16:9 F; 7 M	Runners. Run: at least 10 miles per week. No injuries in the last 6 months.	22 ± 2.4 F; 24 ± 4.5 M
[64]	2018	16:9 F; 7 M	Healthy recreational runners. Run: At least 12 km per week for at min. one year. Experience running 3.2 ± 0.9 years. No injuries in the last year, PPA> 8 g.	25 ± 7.9
[73]	2018	30:12 F; 17 M CG: 20, IG: 10	Healthy runners (late preparation phase for a 5–15 km race). Run: 409 ± 182 min weekly. 11.3 ± 7.5 years of running experience.	31.0 ± 7.5
[71]	2018	16:5 F; 11 M	Healthy recreational runners. Run: at least 5 km per week for about a year. No injuries.	28 ± 6.2
[74]	2016	12:4 F; 8 M	Healthy recreational runners. Run: Min preferred treadmill running speed of 2.3 m/s at least twice a week. No injuries in the last year.	29.67 ± 4.4
[69]	2016	22 M; CG: 11; G: 11	Healthy runners. Run: more than 10 km per week.	18–45
[72]	2016	30:16 F; 14 M 14/16 IG	Healthy runners. Run: at least 11.3 km/week, at-risk runners who exhibited high-impact forces No injuries in the last 90 days. High-impact runners with VILR ≥85 body weight/s in either limb.	22.1 ± 10
[62]	2016	30:14 F; 16 M; 15 IG	Healthy runners, Run: at least 10 km per week, No injuries in the last 6 months, Vertical GRF ≥ 85 body weight/s in either limb.	18–35
[65]	2014	22 M; 12 IG	Healthy runners. Run: at least 30 km per week. No injuries in the last 6 months. PPA >9 g.	CG 33.3 ± 9.0 IG 30 ± 11
[70]	2014	9:6 F; 3 M	Healthy runners. Run: at least 10 miles per week, 2 times per week. Heel-strike footfall pattern.	20 ± 1.5
[66]	2011	10:6 F; 4 M	Healthy runners, PPA tibia > 8 g, Run: 16 km per week. Rearfoot strike running pattern.	26 ± 7
[76]	2011	18:7 F; 11 M	Healthy runners: 16 runners oriented at a national level, 2 recently retired competing at international level. Run: Comfortable running at 16 km/h.	28.4 ± 6.4
[68]	2010	5 F	Healthy runners. Experienced running on a treadmill. Run: at least 32 km per week for at least 3 months.	26 ± 2
[77]	2010	8	Amateur team-sport athletes. Prolonged High-Intensity Intermittent Exercise training time: 2 h, 3 times per week + 1 weekly match. Competed at least during the last 3 years prior to the experiment.	20.6 ± 1.2

Definitions: Female (F); male (M); years (Yrs); control group (CG); intervention group (IG); peak positive acceleration (PPA), ground reaction forces (GRF).

**Table 3 sensors-20-06637-t003:** Equipment, body placement and feedback form.

ID	Sensors and Equipment	Tracking Characteristics	Body Place of Sensors	Type of Feedback and Instruction	Feedback Body Placement and Form
[62]	3D accelerometer and running computer (Forerunner 70™ Garmin) AE: CMCY and an instrumented treadmill	Step rate transmitted to wrist computer	Right shoelace of the shoe	Visual—maintain step rate increment of 75%	Wrist: numerical
[71]	3D accelerometer (Maestro WB, 01DB-Stell) AE: instrumented treadmill	Device tracks tibia acceleration	Posterior heel counter of the shoe	Visual—land softer aiming for a target with a second task	Monitor: colored circles representing target accuracy
[66]	Accelerometer #axis: not mentioned (PCB Piezotronics—Model 356A32) AE: CMCY	Device tracks tibia acceleration	Left anteromedial distal aspect of tibia	Visual—run softer and keep acceleration peaks below the threshold (line)	Monitor: acceleration graph
[68]	Uniaxial accelerometer (PCB Piezotron-ics, Inc) AE: force transducers	Device tracks tibia acceleration	Right anteromedial distal aspect of tibia	Visual—run softer and keep acceleration peaks below the threshold (line)	Monitor: acceleration graph
[69]	3D accelerometer (PCB Piezotronics- Model: PCB356A32/NC) AE: CMCY	Device tracks tibia acceleration	Right anteromedial distal aspect of tibia	Visual/auditory Run softer and keep inside the threshold area (Line?)	Monitor: acceleration graph and clinicians verbal instructions
[61]	2D accelerometer (ADXL278) and MEMS AE: instrumented treadmill	Device tracks acceleration from the shoe and the tibia; Both values represent tibia acceleration	Right anteromedial distal aspect of tibia and posterior heel counter of the shoe	Visual—run softer and keep acceleration peaks below the threshold (green light)	Monitor: dichotomic signal Red and green circle light signals
[70]	3D accelerometer: (G-Link -LXRS: Lord Micro strain) AE: non-instrumented treadmill	Device tracks tibia acceleration	Left anteromedial aspect of the distal end of the tibia	Auditory—avoid sounds	Beep with different pitch levels
[67]	IMU with 3D accelerometer: (YEI 3-space sensor, YEI Corporation AE: unspecified treadmill	Device tracks tibia acceleration	Anteromedial aspect of the distal end of the tibia (unknown leg)	Visual—run softer and keep acceleration peaks below the threshold (line)	Monitor: acceleration graph
[76]	Position transducer (Mod 1850-050, HIS-Houston Scientific International Inc)	Waist strap on the back connected to position transducer Tracking runner center of vertical mass displacement	Waist	Auditory—pre-recorded verbal instructions Visual—reach VD and speed target	Wireless headset: volume proportional to the error Monitor: 3 vertical bars
[75]	Sock embedded pressure sensors (Sensoria^®^, Sensoria Inc) and heel FRS AE: Flexible insole with 99 capacitive sensors (Pedar^®^-X; Novel Inc), treadmill and tablet	Device tracks cadence and identifies if the heel is loaded	FRS placed at the heel of each foot pressure sensors at heel and metatarsals	Visual/auditory—increase cadence, avoid landing on the heel and sound (Red circle?)	Tablet: graphical foot strike Beep (not from iPad) at heel strike
[64]	3D accelerometer (Maestro WB, 01DB-Stell) AE: instrumented treadmill	Device tracks acceleration from the back of the shoe Values represent tibia acceleration	Right anteromedial distal aspect of tibia and posterior heel counter of the shoe	Auditory—run softer and avoid sound Simultaneous cognitive task	High pitched sound representing error intensity proportional to error
[72]	3D accelerometer and wrist computer (Garmin FR70™ and footpad) AE: CMCY and instrumented treadmill	The footpod calculates step rate that is sent to the wrist computer	Right shoelace of the shoe	Visual—maintain step rate increment of 75%	Wrist: numerical Participant free to decide when to check their data
[56]	Accelerometers (model 353B17, PCB Piezotronics accelerometers) # axis not defined, AE: CMCY and mouthpiece Vo2	Device tracks tibia acceleration, and results were displayed as leg and head power spectral densities (PSD)	Right anteromedial distal aspect of tibia and head frontal bone	Visual—maintain a target eversion angle Reference angle from an irregular terrain	Monitor: gray area represents target angle
[73]	3D inertial measurement unit (Axiamote, motion Tracking™ device) AE: HR chest strap, instrumented treadmill, capillary blood sampling	Device tracks step frequency and CGT	Right shoelace of the shoe	Visual—reduce CGT while running with highest possible self-paced	Tablet screen: CGT bar displayed during resting times
[63]	2D accelerometers AE: instrumented treadmill	Device tracks tibia acceleration	Bilateral anteromedial distal aspect of tibia	Visual—land softer	Monitor: acceleration graph
[60]	Not wearable sensors AE: CMCY	Lower body kinematics and COM data were collected with no wearable devices	None	Haptic—run normally while maintaining a preset target speed	Permanent resistance in lower limbs
[74]	Two 3D accelerometers (Trigno; Delsys, Inc) AE: CMCY	Device tracks tibial and head acceleration	Right anteromedial distal aspect of tibia and head frontal bone	Visual—match footfalls to metronome and maintain head in target area (dot line and square box)	12 × 16 m Screen: head trajectory graph and target gaze area
[77]	No wearable sensors AE: nonmotorized treadmill, reflective object sensor and capillary blood sampling	Device tracks running velocity	None	Haptic, visual and audio Keep target speeds and cover maximum possible distance	Compression by a full bodysuit Monitor: numerical speed Audio: tone and verbal command before each run
[65]	3D accelerometer (2400 T G2, Noraxon) AE: infrared CMCY	Device tracks tibia acceleration	Right anteromedial distal aspect of tibia	Visual and auditory Remain under shock threshold with a self-selected running strategy	Large 5 m screen: colored circles representing target accuracy Sound: laboratory speakers

Definitions: Additional equipment (AE); wearable device (WD); heart rate (HR); contact ground time (CGT); force resistive sensor (FRS); camera motion capture system (CMCY).

**Table 4 sensors-20-06637-t004:** Protocol and parameter that triggers feedback signals.

ID	R	Protocol	Follow-up	Trigger
[65]	Yes	6 sessions 20 min each. Feedback every 5 gait cycles all sessions.	1 month	50% from BS PPA. Displayed 50–75% and over 75%
[66]	No	8 sessions in a 2-week period. Increasing 15 to 30 min. Feedback was reduced gradually during the last 4 sessions.	1 month	50% of BS of PPA
[68]	No	Single session of 30 min. 10 min No feedback followed by 10 min feedback.	No	50% of BS of PPA
[69]	No	2 sessions. 30 min. 10 min No feedback followed by 10 min feedback.	1 week	50% of BS of PPA
[71]	No	8 sessions in a 2-week period. Increasing 15 to 30 min. Feedback was reduced gradually during the last 4 sessions. Each condition included a numeral cognitive task. Feedback was randomized.	No	80% of BS of PPA
[61]	No	8 sessions in a 2-week period. Increasing 15 to 30 min. Feedback was reduced gradually during the last 4 sessions.	No	80% of BS of PPA
[70]	No	Single session. 20 min with intermittent feedback every 5 min.	No	75–80% of BS of PPA
[64]	No	8 sessions in a 2-week period. Increasing 15 to 30 min. Feedback was reduced gradually during the last 4 sessions. Each condition included a numerical cognitive task.	No	80% of BS of PPA with a constant speed and cognitive and verbal task
[67]	Yes	9 sessions. 6 min of feedback during the first 8 sessions. Participants had 1-hour endurance training before the feedback. Session #9 was a 1-month follow-up.	1 month	100%, 120% Speed and maintaining PPA under 100% of BS
[63]	No	8 sessions in a 2-week period. Increasing 15 to 30 min. Feedback was reduced gradually during the last 4 sessions.	No	90% and 110% BS speed and 80% of BS of PPA
[73]	Yes	8 outdoor sessions in a 4-week period. Training sessions with a structure mixing 400 m to 1000 m runs. Data collection was measured at 80% of the max speed with 1% of treadmill inclination. Feedback was presented during resting times (approx. 3 min).	1 week	Decrease GCT from previous run with a structured distance protocol aiming 80% of maximal velocity
[72]	Yes	8 outdoor sessions in a 4-week period. Constant feedback during the session. Self-controlled. Run time: free.	1 month	7.5% increment of SF
[62]	Yes	8 sessions (indoor or outdoor) in a 4-week period. Feedback on sessions #1–3, 5 and 7. Constant feedback during the session. Self-controlled. Run time: free. Groups were blinded.	1 month	7.5% increment of SF
[74]	No	“Series of short runs”. Protocol does not specify the frequency or intensity. Feedback session had continuously head trajectory.	No	80%, 90%, 110% speed; 80% to 120% SF
[75]	No	Single run following a metronome. 15 min of feedback. 5 min for each condition. Visual, auditory and combined.	No	Incremented 10% BS of cadence and if heel strike load was sensed
[76]	No	Single session. Running or feedback time not specified. 11 running trials combining 2 control variables and feedback: Vertical displacement and step frequency.	No	80%,90% and 100% VD; 80% and 90% of BS SF
[56]	Yes	Single session. 30 min session. 10 min control, 10 min irregular surface, 10 min with feedback.	No	Matching FA of irregular surface run ±1 degree
[77]	Yes	Single session. 45 min high-intensity session. 3 runs of 15 min with 6 different running speeds and last 2 min with self-selected speed covering as much distance as they can. Continuous feedback.	No	0%, 20%, 35%, 50%, 70% and 100% of BS sprint. 50% considered running
[60]	No	Single session. 6 min control and 6 min feedback with 10 min rest. 6 min counted from reaching self-selected speed with typical moderate intensity.	No	5 kg resistance in lower limbs from a wearable and portable belt

Definitions: Randomized (R); baseline (BS); positive peak acceleration (PPA); vertical displacement (VD); step frequency (SF).

**Table 5 sensors-20-06637-t005:** Biomechanics changes produced by articles that provided feedback based on PPA.

ID	Parameter of Change	PPA	VALR	VILR	VIP	Ankle Angle	Comments
[65]	PPA 50%	31% R: 22%	18% R: 6%	19% R: 9%		↓ 200%; ↓ FStrike angle 44%	R: 1 month
[66]	PPA 50%	48% R: 44%	32% R: 27%	34% R: 30%			R: 1 month
[68]	PPA 50%	17–60%	16–39%	15–39%	10–30%		Results were given as a tendency. Values were analyzed in each participant.
[69]	PPA 50%	18.9%; R: 21.2%					R: 1 week
[71]	PPA 20% (cognitive load)	41%	24%	18%			
[61]	PPA 20%	26.4%*	16.40%	17.30%			* Value from the tibia. PPA in the shoe: 40.9%. However, PPA in the shoe was not significantly correlated with VALR and VILR. The measurements registered in the shoe were 4 times higher than in the tibia.
[70]	PPA 15–20%	11%–8%					Values changed across checking points over time—starting in 11% ending in 8%.
[64]	PPA 20% (cognitive load)	33.8%*	25%	22%			* Values from the shoe. PPA in the tibia: 10%; post feedback: 21.5%, but there was no significant correlation with VALR and VILR.
[67]	↑Speed 120% and free ↓ PPA	16%; R: 16%					R: 1 month. Significant changes observed from the 5th session. Complementary tables are not given. Retention: 1 month.
[63]	↑&↓Speed 10% and ↓ PPA 20%	R: 35–37%					R: 1 week. Changes observed across all speeds. There was no significant difference between limbs. Training effect did not interact with speed.

Definitions: Decreased (↓); retention (R); foot strike angle–angle between the floor and the foot (FStrike), (*) details about a value in comments.

**Table 6 sensors-20-06637-t006:** Biomechanics changes produced by articles that provided feedback based on SF.

ID	Parameter of Change	SF	VD- COP	Ankle-Foot Changes	Knee Changes	Hip Changes	Spatiotemporal Changes	Comments
[62]	SF ↑ 7.5%	↑ 8.6%; R: 8.5%	↓: 9% *; R: 6.2%	kinematics: ↓Foot at FStrike: 22%, R: 19% ↓ Tibia at Fstrike: 31%; R 35%	kinematics: ↑ Flex at Fstrike: 25%; R 32%. kinetics: CKF: ↓ 7.6–10.61% R: 4.5–8.24%.		Step length: ↓ 8.36%; R: 8.2%	* COP distance from body COM at FStrike
[72]	SF ↑ 7.5%	↑ 8.6%; R ↑ 8.5%		kinetics: VALR; 19%; R: 20%; VILR: 18%; R: 17.5%	kinetics: Abs↓ 27.27% R: ↓ 21.21%; Abs/km ↓ 21.1%, R: ↓ 14%	kinematics: Add ↓ 27%; R: ↓ 21%		
[75]	SF ↑ 10% and loading FS	↑ 10%	COP foot position: Moved forward SF: 6%; SF and Visual FB: 37.31%; SF and Auditory FB: 45.35% Combined: 45.51%	kinetics: Area loaded: midfoot ↓ 40%, heel ↓ 80%, forefoot ↑ 10–36%; great toe ↑ 30%				
[74]	SF (↑ 10%, 20%, ↓ 10%, ↓ 20) & head trajectory	↑ 10–20%		kinetics: Tibia shock attenuation: SF + 20 = ↑ 12%; SF + 10 = ↑ 5% peak impact acceleration ↓ 1.9%				Significant changes across all SF conditions. Larger impact on Tibia with SF-20. Changes reported in this table: +20, +10. Feedback was not significantly correlated with variables.
[76]	SF ↓ 0–10%; VD: ↓ 10–20% and P ↓ 10–20%	↑ up 7% and ↓ 10% *	↓ 20% and ↑ 5% *					Trend value. Not significant

Definitions: Decreased (↓), increased (↑), foot strike (FStrike); absorption (Abs); adduction (add); 1-month retention (R); run power (P); step frequency (SF); feedback (FB). (*) details about a value in comments.

**Table 7 sensors-20-06637-t007:** Biomechanics changes produced by articles that provided feedback based on uncommon variables.

ID	Parameter of Change	SF	Ankle–Foot Changes	Knee Changes	Spatiotemporal Changes	Metabolic	Comments
[73]	CGT ↓ defined by the user	1.5↓			↑ swing time: 2.32%; ↓ run time (400 m):1.5%; ↓ TOGC: 1.9%	↓ HR: 2.5%	
[56]	↓ Inv to BS irregular surface		kinematic: Inv ↓ 20% (↓ 14.4) * kinetic: PSD: ↑ 35% *	Flex at Fstrike: ↓ 20.7% (↑ 17%) *		↓ VO2:10%.	* Inv and knee flex ↓, but it was tested on a different surface. On the same surface, they ↓ 14.4% and ↑ 17%, respectively, but with no significant differences. * SPD ↑, but shock attenuation was the same (−9.8).
[77]	50% of BS sprint speed				↑ Distance coved: 7.1%; ↑ Speed: 8.5%;	↑ HR 1–2%; 4% ↑ TOI;	Lactate was not significant
[60]	Device resistance (5 k)		VD ↓ 12.7%	kinematic: Angle of tibia at Fstrike: ↓ 34.7%, ↓ Exc: 14.9%; ↑ Flex: 6%			Step length: no significant changes.

Defintions: Decreased (↓), increased (↑), contact ground time (CGT:); total ground contract time (TOGC); inversion (Inv); base line (BS); tissue oxygenation index (TOI); power spectral density (PSD), heart rate (HR), vol max O2 (VO2), (*) details about a value in comments.

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
