# Peer review of "Effects of Wearable Devices with Biofeedback on Biomechanical Performance of Running—A Systematic Review"

_sensors, 2020, doi:10.3390/s20226637_

Round 1
Reviewer 1 Report
The thesis statement in the conclusion is evident:
"The purpose of the current review is to determine if wearable devices with biofeedback increase running performance."
However, as I read the rest of paragraphs, they are vaguely supporting this thesis statement. I would personally suggest that each paragraph should support a response to this particular statement. That is, is there supporting evidence that wearable devices are improving running performance? If yes, what is lacking in the literature. If not, what areas need to be investigated further. These point may have been covered by the authors but they might be so implicit which will give the readers hard time to conclude anything.
I think a restructuring the conclusion with a focus on addressing the research question in the paper may prove helpful.
Author Response
Point 1: The thesis statement in the conclusion is evident: "The purpose of the current review is to determine if wearable devices with biofeedback increase running performance." However, as I read the rest of paragraphs, they are vaguely supporting this thesis statement. I would personally suggest that each paragraph should support a response to this particular statement. That is, is there supporting evidence that wearable devices are improving running performance? If yes, what is lacking in the literature. If not, what areas need to be investigated further. These point may have been covered by the authors but they might be so implicit which will give the readers hard time to conclude anything. I think a restructuring the conclusion with a focus on addressing the research question in the paper may prove helpful
Response 1:
Conclusions were revised and statements added supporting the thesis. Not relevant information was removed.
Lines 552-545:
“positive peak acceleration, which was found to be associated with a tibial stress fracture, was significantly reduced upon receiving its biofeedback. By providing biofeedback to increase stride frequency, participants had significant reductions in the knee and ankle vertical load and a significant increase in knee flexion. Positive changes were also found in wearable devices which fed back other parameters (contact ground time and speed), as they reduced running time, vertical displacement and increased greater knee flexion”
Lines 549-551:
"In addition, more randomized-controlled trials which are conducted in outdoor environments are required to better evaluate the effects of wearable devices on runners with different levels of runners"
Reviewer 2 Report
- The abstract section needs significant modification. It should be focused on reviewing the specific performance parameters and the enhancement needs to be mentioned very briefly in the abstract.
- Major layout modification would improve the manuscript.
- What other parameters can be involved in running performance rather than the paper used as the dataset?
- What possible error factor could be involved in analyzing the performance of wearable devices?
- In the conclusion "The lack of 565 consistency across the measurement of the variables that define running performance and injury 566 prevention challenges the capacity to provide conclusive evidence. " is mentioned. This needs a more detailed analysis in the discussion section.
Author Response
Point 1: The abstract section needs significant modification. It should be focused on reviewing the specific performance parameters and the enhancement needs to be mentioned very briefly in the abstract.
Answer 1: The abstract was modified, the following statements were added to focus on the performance parameters: (lines 26-34)
“Most reviewed articles reported a significant reduction in positive peak acceleration, which was found to be related to tibial stress fractures in running. Some previous studies provided biofeedback aiming to increase stride frequencies. They produced some positive effects on running, as they reduced vertical load in knee and ankle joints and vertical displacement of the body and increased knee flexion. Some other parameters, including contact ground time and speed, were fed back by wearable devices for running. Such devices reduced running time and increased swing phase time. This article reviews challenges in this area and suggests future studies can evaluate the long-term effects in running biomechanics produced by wearable devices with biofeedback”
Point 2: Major layout modification would improve the manuscript.
Answer 2:
-Table 1 was included to visualise better the strategy search. Page 3.
-Table 3 (before 2) was modified significantly to better summarize key points. Page 5.
-Figure 2 (page 7) was modified to represent better the studied wearable devices and paragraph 3 page 7, lines 165-168 were reviewed.
-Introduction was revised and modified. Paragraphs were restructured: lines 39-49, line 69, 74, 75-82.
-Paragraphs in the Discussion were reviewed: lines 394-399; 402-409; 462-472; 499-501.
-A new section – “Future research direction” was included to highlight the areas that need further research: lines 506-517.
-Conclusions were modified to narrow the biomechanical parameters related to running performance. Lines 537-643; 548-550
Point 3: What other parameters can be involved in running performance rather than the paper used as the dataset?
Answer 3: Biomechanics running performance parameters were added on page 21, lines 463-467:
“Better running performance has been associated with lower percentage of stance phase, smaller peak breaking velocity of the pelvis, greater upright trunk [90], slower velocity in the knee during stance phase, faster knee flexion velocity during swing phase as well as faster hip flexion velocity between heel strike from opposite feet [90-92]. These parameters were not involved in any of the articles analyzed”.
Point 4: What possible error factor could be involved in analyzing the performance of wearable devices?
Answer 4: Possible error factors were added on page 02, lines 75-79:
” Wearable sensors were validated against gait laboratory equipment on kinematic and kinetic measurements [48-54] in sports-related movements [55]. Accuracies of wearable sensors varied, as there were always challenges in accurate position of sensors [56], data processing and simplification methods in calculating joint angles and segment acceleration [57] and protocols to the devices [58]”.
Point 5: In the conclusion "The lack of 565 consistency across the measurement of the variables that define running performance and injury 566 prevention challenges the capacity to provide conclusive evidence. " is mentioned. This needs a more detailed analysis in the discussion section.
Answer 5: The statement was deleted, conclusions and discussion were revised.
Reviewer 3 Report
The performed review work regarding Effects of Wearable Devices with Biofeedback on Biomechanical Performance is good, I have a few minor recommendations for the authors:
- In the second line of the introduction, maybe authors can add a 1-line description of what they are referring to as the term 'Biomechanics'. Maybe one-two descriptions of a running process of a human in terms of knee flexion, hip extension, or plantarflexion would be wonderful for a new reader. An image might serve better.
- I would also add a figure of IMU placement for the third paragraph of Page 7 (line 179-182).
- I would also replace line 112-127 with a table. The description of 112-127 is not properly accessible.
- Table 2 is too large to consume. I recommend to include only important key points in the table. The interested reader would definitely open up those papers if required, No need to include all information from all paper. You should make Table 2 one fourth of its current form.
- If possible, I would provide another section as a 'future research direction'. That information is hidden inside the large discussion section.
Author Response
Point 1: In the second line of the introduction, maybe authors can add a 1-line description of what they are referring to as the term 'Biomechanics'.
Answer 1: Meaning of biomechanics was included in page 1, lines 44-45:
“Attempts to change the biomechanical parameters of runners, characterized by quantitative spatiotemporal, kinematic and kinetic data [12,13],
Point 2: Maybe one-two descriptions of a running process of a human in terms of knee flexion, hip extension, or plantarflexion would be wonderful for a new reader. An image might serve better.
Answer 2: The running process was generally described using three descriptions in hip, knee and ankle respectively: Page 1, lines 39-43:
“During running, while the hip generates power to accelerate the leg so as to optimize the position of the foot and center of body mass [3-6], the ankle stabilizes and further accelerates the limb forwards [3,7,8] and the knee absorbs the loading by increasing the muscular power [3,9,10]”.
Point 3: I would also add a figure of IMU placement for the third paragraph of Page 7 (line 179-182).
Answer 3: References were reviewed, the paragraph was changed as (lines 165-168):
“The most popular body place among studies that used accelerometers was in the anteromedial aspect of the distal the tibia, approximately 15cm above the medial malleolus”
Figure 2 was modified. The IMU are represented as boxes, the same as the accelerometers. The figure was included on page 7.
Point 4: I would also replace line 112-127 with a table. The description of 112-127 is not properly accessible.
Answer 4: Table has been added (table 1 – page 3), line 113-115 was included and the paragraph with the search strategy was removed.
Point 5: Table 2 is too large to consume. I recommend to include only important key points in the table. The interested reader would definitely open up those papers if required, No need to include all information from all paper. You should make Table 2 one fourth of its current form.
Answer 5: Table 2, now table 3 was reduced from 7 pages to 3 pages (page 8).
Point 6: If possible, I would provide another section as a 'future research direction'. That information is hidden inside the large discussion section.
Answer 6: A section called 'future research direction’ has been added: lines 506-517
“Future research direction: Further research in the running efficiency and economy upon using wearable devices with biofeedback would be valuable. Wearable devices are a powerful tool to modify the running technique, yet, laboratory equipment was used to evaluate the effectiveness of wearable devices. Further research on the validation and calibration of these devices would be important to increase the number of interventions in the field and to fully understand the effects of such technology in running outdoor.
Long term effect upon interventions with wearable devices is an unexplored area. Research has focused on short terms effects upon short interventions, however, the evidence highlights promising long term effect on the positive peak acceleration in those protocols where the subjects were exposed to longer periods of time and the task did not focus on their metrics. It would be beneficial to increase the number of clinical trials exploring the long term effects of wearable devices with multiple biofeedback modalities in biomechanical parameters”